# Scale Effect on Impact Performance of Unidirectional Glass Fiber Reinforced Epoxy Composite Laminates

**DOI:** 10.3390/ma12081319

**Published:** 2019-04-23

**Authors:** Yiou Shen, Bing Jiang, Yan Li

**Affiliations:** 1School of Aerospace Engineering and Applied Mechanics, Tongji University, 1239 Siping Road, Shanghai 200092, China; syoeva@tongji.edu.cn (Y.S.); jiangbing@163.com (B.J.); 2Institute of Engineering Mechanics, 29 Xuefu Road, Ha’erbin 150080, China; 3Key Laboratory of Advanced Civil Engineering Materials, Ministry of Education, Tongji University, Shanghai 200092, China

**Keywords:** scale effect, low-velocity impact, fiber reinforced composite, impact damage

## Abstract

As a result of the increasing use of glass fiber reinforced plastic (GFRP) composites in engineering fields, the investigation of scale effect on impact performance for this kind of composite is essential for large scale structure design. The effects of scaling on the impact response of simply supported unidirectional GFRP were investigated through drop weight impact (DWI) tests in this study. Impact tests were undertaken over a wide range of energies to generate damages between barely visible and initiated penetration on four scale size GFRP laminates. The main impact responses including impact force, contact duration, displacement, energy absorption and damage area of scaled specimens were normalized to compare with the full-size specimen. It was found that the impact response of large sample with elastic deformation and small area of delamination can be predicted accurately according to a geometrical similar scaling law. Scale effect was found in the damage threshold force and absorbed energy of the laminates when significant internal damage occurs due to the microstructural effect becoming important in resisting impact force and absorbing impact energy. Moreover, the energy partition and effective stiffness were calculated according to the energy balance model to reveal the contribution of different modes of deformations on energy absorption for the GFRP laminates.

## 1. Introduction

As fiber reinforced composite materials are being more widely used in the aerospace industry, and fields like construction and yacht hulls, an on-going constant source of concern is the effect of foreign object impacts on their mechanical properties [1,2]. Impact is a dynamic event, large deflections and membrane effects are usually significant during impact, which cause numerous damage modes, such as matrix cracking, delamination, and fiber fracture and so on. The impact behavior and damage modes of composite structure are closely related to the inherent character of the composites, as well as their dimensional scales [3]. Hence, due to the overall complexity of the impact problem, experimental data are needed to determine the impact response and resulting damage modes in particular material systems and structural geometries. For economic reasons, impact experiments are frequently carried out at a laboratory scale. It is necessary to be able to scale this response to real structures of interest [4,5,6].

Scaled tests are a reliable way to investigate size effects of composite materials, if the complete test arrangement is correctly scaled, the stress distribution should be the same in all samples [7]. Even if there are stress concentrations, they should have the same effect on the all sized specimens. This requires the complete test geometry to be scaled, including any loading fixtures [8]. In recent years, many experimental studies have been conducted on the mechanical scaling behavior of composites. Most studies have used two main approaches for scaling composites [9,10,11,12,13,14]. The first is ply-level scaling, which involves simply increasing the number of layers for each angular ply orientation. The second approach, called sublaminate-level scaling, involves increasing the thickness of the laminate by repeating the baseline stacking sequence as a sublaminate group. 

The difficulty in predicting the behavior of impact damage and failure with changes in scale is that the underlying failure mechanisms are not well understood [15]. Swanson [16] studied the scaling of impact damage in carbon/epoxy plates from laboratory specimens to industrial structures. He found that the delamination exhibits a dependence on the specimen’s size. However, the geometry dimension does not have a significant effect on the failure stresses or strain under impact loading. Sutherland and Soares [17] used the dimensional analysis approach to develop scaling laws for impact on fully clamped low volume fraction woven roving and short E-glass fiber composite materials. They revealed that the impact responses for this type of composite also scaled well for the elastic response. Viot et al. [18] verified their findings on unidirectional carbon/epoxy composites. Oshiro and Alves [19] used dimensionless and mathematical models to calculate a correction factor for the impact velocity when scaling impacted structures. They only considered strain rate effects and observed the fact that for structures under impact loads, the scaling laws maybe distorted. Yang et al. [20] investigated scaling effects in a sandwich structure based on a carbon fiber reinforced composite (CFRP) skin and a polymer foam core. The size effects in the load–displacement responses, the total absorbed energy as well as the resulting damage within the sandwich structures were focused on and the simple scaling laws found can be used to predict the low velocity impact response of sandwich panels. Many studies have shown the high degree of equivalence between quasi-static indentation and low-velocity impact damage in laminated composites [21,22]. For this reason, some researchers used static indentation tests to qualitatively elucidate the damage induced during low-velocity impact events. Bogenfeld et al. [23] delivered an analytical scaling approach for low-velocity impact on composites based on the spring-mass model. Their approach permits the analysis of structural impact scenarios based on a single coupon simulation to achieve a real impact assessment of a large structure. Later, they conducted a series of impact tests to assess the analytical scaling method. They identified higher-order vibration as playing an important role in affecting the energy balance of impacts on the structural level [24]. Abisset et al. [25] found that all scaled composites laminates during the static indentation test exhibited similar damage patterns despite being loaded at different levels. They found the in-plane dimension is the most effective parameter to be scaled to reduce the size of the tested specimens, thereby reducing testing and computational costs. Wagih et al. [26] proposed an analytical model to predict the impact response of large structures by using the static indentation test results of small-scale samples. It can be concluded that the available scaling laws are only valid to scale the elastic response of the maximum impact force and sometimes the delamination area. However, there is a lack of knowledge in scaling of other impact responses such as contact duration, displacement, damage size and the absorbed energy for a wide range of impact masses ranges from similar to the sample to over ten times of the sample.

In this paper, the aim is to investigate the scaling effects on impact response in a unidirectional glass fiber reinforced epoxy resin composite. The geometrical dimension effect on impact response including impact force, contact duration, displacement, and energy absorption as well as the resulting damage within the glass fiber reinforced plastic (GFRP) laminates are detailed analyzed. A geometry similar scaling law is used in this work for assessing the possibility of using scale models for predicting the full-scale impact response.

## 2. Scaling Theory

The scaling effects on the low-velocity impact of GFRP laminates are investigated by using a geometry similar scaling law developed by Morton [4]. This approach is strictly a dimensional analysis method, using the well-known Buckingham’s П-theorem. It involves 13 input and response parameters as shown in Table 1. According to the Buckingham’s П-theorem, there are the following 12 independent П terms:

Test parameter: $\prod_{1}=\delta /h$

Geometric: $\prod_{2}=R/h$, $\prod_{3}=R_{i}/h$

Material property: $\prod_{4}=\upsilon$, $\prod_{5}=\upsilon$, $\prod_{6}=\rho /\rho_{i}$, $\prod_{7}=E_{i}/E$

Impact event: $\prod_{8}=\rho_{i}{V_{i}}^{2}/E$, $\prod_{9}=m_{i}/h^{3}\rho_{i}$, $\prod_{10}=tV_{i}/h$, $\prod_{11}=P/h^{2}E$, $\prod_{12}=m_{i}{V_{i}}^{2}/2h^{3}E$

In this study, the dimension of the specimen, impactor and support ring diameter were scaled as “n”, the impactor mass and the impact energies were scaled by “n^3^”, the resulting contact force and damage area were scaled by “n^2^”. Other parameters showed in Table 1 with scale factor of 1 are not scaled in this investigation, and these values are constant for all the tests. 

## 3. Materials and Methods

### 3.1. Materials and Fabrication

The glass fiber reinforced epoxy composite investigated in this research was based on unidirectional E-glass fiber reinforced FM94 epoxy resin prepreg supplied by the Advanced Composites Group Ltd., East Midlands, UK. The mechanical properties of the prepreg system and the fabrication process is detailed in previous work [27]. The stacked sequence of the GFRP laminates is [0,90]_λs_ (λ = 1,2,3,4) and the resulting fiber volume fractions are all 55%. All of the panels were subjected to the same manufacturing conditions, i.e., time, temperature and pressure, to avoid unwanted variations in the laminates. After manufacturing, the square panels were cut using a slitting wheel and finished to achieve the appropriate scale size. Four scaled specimen sizes with values of the scale factor n corresponding to n = ¼, ½, ¾ and 1 (full-scale size) were manufactured to investigate the scale effect of the GFRP laminates. The geometrical dimension of the GFRP specimens was scaled as shown in Table 2. 

### 3.2. Low-Velocity Impact Test

The low-velocity impact response of the scaled GFRP laminates was investigated using an Instron CEAST 9350 drop-weight impact machine (Instron, Norwood, MA, USA) at room temperature. Here, hemispherical indenters were also geometrically scaled with diameter of 20n mm (5, 10, 15 and 20 mm), and were used for the four scale size of specimens, respectively. These four scale size samples were simply supported by four scaled steel circular support rings as shown in Figure 1, the circular support has an inner diameter of 200n mm (50, 100, 150 and 200 mm), as listed in Table 2. A release height of 500 mm was employed for all specimens to achieve a constant impact velocity of approximately 3 m/s. Impact force data were acquired as voltage output and then transferred into a module 64K DAS (Data Acquisition Station) at a frequency of 100 kHz. Impact velocity was acquired by a photoelectric sensor. During the impact test, the impactor was released and dropped vertically passing through the photoelectric sensor beam, and the impact velocity was detected when the tip of the impactor just touched the surface of the specimen. The error of the measured velocity is within 0.01 m/s. The displacement was calculated by Pro Analyst software (Workstation), basically considered from load–time relation. The friction and heat transformation between impactor and samples were neglected and the incident energy was assumed be absorbed by the sample. In order to predict the full-scale size impact response of the specimens, a range of initial impact tests was conducted with impact energies ranging from relatively low values, which produce large elastic or plastic deformations in the samples up to the high values which cause matrix cracking, delamination and penetration in the n = ¼ scale size panels. After initial tests were carried out, the masses of the carriage (hence the impact energy) used for all size specimens were determined accordingly. It was found in the previous work [3] that the impact force of this type of GFRP composite laminates is reproducible. Therefore, only one sample was tested for each impact energy in this study. The drop weight impact (DWI) tests were undertaken over a wide range of energies to generate damage between that which is barely visible and large-scale delamination as well as initiated penetration. The impact mass and energy applied on each test were given in Table 3.

## 4. Results

### 4.1. Scale Effect on Impact Force and Duration

Figure 2 shows the typical impact force versus time histories for four scale size specimens tested at a range of energies, including 11.93n^3^, 50.26n^3^, 69.68n^3^, 99.32n^3^, 120.78n^3^ and 150.04n^3^ J. It can be clearly seen that increasing the impact energy leads to the enhancement of impact force and contact duration for all size samples. As expected, the maximum impact forces are strongly dependent on the incident energy. These different levels of incident energy caused distinct impact responses for GFRP laminates. The force-time traces show similar trends for all four scaled size samples following lower incident energies of 11.93n^3^, 50.26n^3^ and 69.68n^3^. For the lowest impact energy of 11.93n^3^, relatively large amplitude of oscillation was found for the force-time trace due to some ringing inside the load cell and dynamic effects in the plate. No evidence of load drops on the force vs. time traces of these three sets of impact energies was found in spite of the fact that delamination has initiated at the impact zone. It was found that the full-scale size GFRP laminates achieved damage threshold at an incident energy of 99.32 J, this indicating significant damage occurred in this laminate after impact. However, other three scale size sample achieved damage threshold at a higher incident energy level of 120.78n^3^ J, which means smaller size samples exhibit better impact resistance for this type of composite. As for the impact energy of 150.04n^3^ J, the maximum force drops sharply after reaching the peak due to the damage propagation. A clear inflection point can be seen in the ½, ¾ and full-size specimens as the impact load decreases.

Scaling effects in the force-time traces were investigated by dividing the impact force by the square of the corresponding scale size, (F/n^2^), and the timescale by the scale size, (t/n). Figure 3 shows the resulting normalized force-time curves. An examination of the first two figures of 11.93n^3^ and 50.26n^3^ J indicate that all of the normalized traces are very similar in appearance, this suggests that the impact forces and durations for this type of composite can be scaled accurately at relatively low impact energy. However, the scaled maximum impact force of ¼ scaled size sample was much lower than the other three scaled size samples following impact energy of 99.32n^3^, 120.78n^3^ and 150.04n^3^ J. The dash line represents the maximum impact force of the full-scale samples at an impact energy of 150.04n^3^ J, with a 30% discrepancy between the ¼ and full-scale samples, as shown in Figure 4a. In contrast, the impact duration for these four size samples were scaled relatively accurate at a whole range of the impact energies, with only a 15% discrepancy found between the values of ¼ and full-scale samples at the highest impact energy, as shown in Figure 4b.

### 4.2. Scale Effect on Impact Displacement

Similarly, the impact force vs. displacement traces of four scale size GFRP laminates following six series of impact energies were scaled as shown in Figure 5. It is evident that all four panels exhibit almost complete recovery for the three series of small impact energy of 11.93n^3^, 50.26n^3^ and 69.68n^3^ J, indicative of an elastic response of the composite and little energy being absorbed by the specimen during impact. In addition, the load-displacement curves are nonlinear up to the maximum deflection. Such non-linearity is likely to be due to membrane stiffening effects at higher displacements. An inspection of these three figures indicate that all of the normalized traces fall onto a single unique trace, suggesting that the GFRP obeys the geometrical similar scaling law at a lower impact energy level. It is worth noting that internal damages already generated in these four size samples despite the impact energy level were relatively low, the damage modes will be discussed in Section 4.3. However, the differences in the maximum forces between ¼ scaled size laminate and other three size laminates following impact energies above 69.68n^3^ J caused the discrepancies on laminate stiffness and energy absorption. The absorbed energy is obtained by the area under the force vs. displacement curve. An examination on normalized force vs. displacement curves of 150.26n^3^ J suggests a different failure mode in the ½ size sample, where there is a steady decrease in load after the peak force. In addition, a permanent residual displacement is evident in the curves for the ½ size sample. Nevertheless, it is evident that there is reasonable agreement between the four sets of scaled displacement as shown in Figure 6a, although the scaled absorbed energy of the ¼ size GFRP laminates following impact energies of 150.26n^3^ J in Figure 6b is 30% lower than the value of the full-scale size sample, the dash line again represents the absorbed energy of full-scale size sample. 

It is assumed that the laminates respond quasi-statically during the DWI event and the kinetic energy of the impactor is mainly absorbed by bending/shear, membrane and contact deformations for simply supported samples at elastic deformation. The energy dissipated through surface friction effect, material damping and sound waves are ignored here. Therefore, the contribution of energy absorbed by these three deformations are calculated through an energy balance model as mentioned in the previous work [27]. It was found that the variation trend of energy partition with impact energy for four different scale size samples are similar. Figure 7 gives the energy partition profiles of the full-scale size GFRP laminates to show the trends of energy absorption. Where, *W_m_*, *W_b/s_* and *W_c_* represent the energy absorbed by the membrane, bending/shear and contact deformation, respectively. It can be seen that the membrane and bending/shear deformation contribute almost equally in energy absorption at a low level of impact energy for elastic deformation. However, when penetration is initiated, the membrane deformation absorbed nearly 80% of the energy, and the energy absorbed by bending/shear deformation decreases dramatically to 10%, which is similar to the contribution of contact deformation. The effective stiffness is calculated to revel the variation of stiffness within different scale size laminates. *K_b/s_* summarized the effective bending and shear stiffness of the GFRP laminate as both the associated forces are proportional to the global indentation, *K_m_* is the membrane stiffness, which represents the membrane stretching within the laminates. The effective bending/shear stiffness *K_b/s_* can be determined from the individual stiffness using [28]:(1)$$K_{b/s}=\frac{K_{b}K_{s}}{K_{b}+K_{s}}$$

The stiffness terms *K_b_* and *K_s_* for the GFRP laminate are bending and shear stiffness, respectively. They are given as [28,29]:(2)$$K_{b}=\frac{4\pi E_{r}h^{3}}{3(3+\nu_{r})(1-\nu_{r})a^{2}}$$
(3)$$K_{s}=\frac{4\pi G_{zr}h}{3}\left(\frac{E_{r}}{E_{r}-4\nu_{zr}G_{zr}}\right)\left(\frac{1}{\frac{4}{3}+\log\frac{a}{a_{c}}}\right)$$
where, *E_r_*, *υ_r_* and *G_zr_* are the Young’s modulus, Poisson’s ratio and shear modulus of the plate respectively. In this study, the *E_r_* = 28.3 GPa, *υ_r_* = 0.3 and *G_zr_* =5.0 GPa, *a* and *h* represent radius and thickness of the laminate, respectively, *a_c_* is the radius of contact between the projectile and the plate and this was assumed to be equal half of the plate thickness. The subscripts element *z* and *r* represent the longitudinal and radial directions of the laminate. The membrane stiffness *K_m_* are given as [28]:(4)$$K_{m}=\frac{\pi E_{r}h}{a^{2}{\left(3+\nu_{r}\right)}^{4}}\left(\frac{191}{648}{\left(1+v_{r}\right)}^{4}+\frac{41}{27}{\left(1+\nu_{r}\right)}^{3}+\frac{32}{9}{\left(1+\nu_{r}\right)}^{2}+\frac{40}{9}(1+v_{r})+\frac{8}{3}\right)$$

It can be seen from Table 4 that the bending/shear stiffness increases when the scale size increases, whereas, the effective membrane stiffness exhibits the opposite trend.

### 4.3. Scale Effect on Damage Modes and Damage Area

Figure 8 shows the back face damage zones of the four scale size plates subjected to impact energies of 11.93n^3^, 50.26n^3^, 99.32n^3^ and 150.26n^3^ J. The images clearly show that increasing the scale size leads to an increase in the damage area in the scaled GFRP plates. Due to the transparent properties of this material, strong back-lighting can be used to view the internal damage. With some care, it was possible to distinguish between the internal delamination and the front and back face damage. An initial examination of the damage zone on all four size laminates following impact energy of 11.93n^3^ and 50.26n^3^ suggests that the type of matrix cracking and delamination are alike for all four size laminates, in spite of the fact that the edge length of the specimens varies from 65 to 260 mm. For samples impact with energy of 11.93n^3^ and 50.26n^3^, the presence of localized delamination is very clear, it was a more diffuse zone of matrix cracking and fiber-matrix debonding. Fibers damage accompanied with matrix cracking were first observed in full-scale size sample incident with 100 J. However, no sign of fiber fracture was observed on other three scale size samples at this level of impact energy, this verified the findings of load drop was only happened on the force-time trace for the full-scaled size laminates. When the impact energy was 150.26n^3^ J, all four scale size samples were penetrated. From the front face images, a permanent dent appears on the surface of all four scale size specimens, indicating that initial penetration has occurred at this impact energy level. Fiber fracture around the point of impact was also evident on the front face, this reflecting as the maximum force drops sharply in the force-time traces due to significant damage propagation. On the back face, the fibers completely pulled out and the last ply was peeled off from the laminate. Similar observations were apparent in all four scaled panels. The internal delamination formed a “diamond” shape in all scaled specimens. 

The translucent nature of the composite plates enabled the damage area to be readily identified and characterized. The damage area was measured by photographing the samples and tracing the perimeter of the damage zone onto standard graph paper. The damage area was plotted against scale size and illustrated in Figure 9a. As expected, the degree of damage increases steadily with increasing scale size n. Previous work on scaling the low velocity impact response of composites has shown that the impact delamination area should scale as n^2^. The normalized damage areas of four size samples following six sets of incident energies all exhibited nearly horizontal lines as shown in Figure 9b, which indicates that the delamination area also followed a geometrically simple scaling law.

## 5. Discussion

It was found in Figure 2 that damage initiation only occurred in the full-scale size sample when the impact energy is 99.32n^3^ J. This indicates that the scale effect existed in damage threshold of this type of GFRP laminates. Wagih [26] also found that an increment of ply thickness reduces the damage threshold load, but it does not influence the load after damage initiation. It is apparent from Figure 4 that increasing the impact energy leads to the maximum force increase for each of the size samples, when the impact energy is below 120.78n^3^ J. However, for the highest impact energy of 150.26n^3^ J, most of the maximum impact forces are similar to those at an impact energy of 120.78n^3^ J, since the damage thresholds were achieved at an impact energy of 120.78n^3^ J. It was found that significant damages including fiber fracture and large area of delamination has already occurred within the plates at impact energy of 120.78n^3^ J. Moreover, the normalized force data of four scale size samples at lower impact energy level are nearly horizontal, indicating that the maximum impact force follows the scaling law due to only matrix cracking and small area of delamination occurred at these three sets of impact energy, this results also verified other researcher’s findings [16,18,25]. This is due to the reason that the process of delamination growth defined by means of linear elastic fracture mechanics. For the remaining three sets of data, the impact forces of ¼ size GFRP laminates are deviated from the horizontal line and the maximum deviation is nearly 30% for the highest impact energy. However, the other three series of scaled data are relative reasonable. It can be seen from Table 4 that the two important bending/shear and membrane effective stiffness varies with the scale size of the samples, the bending stiffness scales as *h*^3^/*a*^2^ and the membrane stiffness scales as *h*/*a*^2^, which means different deformation manners play different roles in resisting impact loads for four scale size samples. Moreover, the damage modes become complicated at moderate and high impact energy, this including indentation, fiber/matrix debonding, fiber fracture, large area delamination and penetration. Therefore, when these significant damages occur in the laminates, the microstructural effect would become more important in the smaller samples, which caused the large scale of deviation on the impact force for the ¼ size sample. The scale results of the contact duration for all four scale size samples at six series of impact energies are relatively reasonable.

The scaled displacement data indicates that the displacement of larger size samples can be accurately predicted by that of the small size sample according to the scaling law with impact energies causing damage ranges between barely visible and penetration initiation. This also suggests that the global deformation degree of four size laminates are in coincidence. Moreover, the observed damage modes and damage area for four size GFRP laminates following six series of impact energies are generally identical. This is due to the reason that the damage area is mainly caused by delamination, which is governed by energy release rates. However, the normalized absorbed energy for ¼ size sample are 30% lower than that of the other three size samples at impact energy of 150.26n^3^ J of which penetration was initiated. Moreover, previous work [27] showed that increasing the diameter of the plate increases its ability to store energy in elastic modes of deformation. Therefore, more energy was absorbed by larger size laminates. However, Yang and Cantwell [3] found the total energy absorbed by the sandwich panels is roughly constant for each scale size, at each impact energy. They suggest that any energy that might be lost in the experimental rig is constant, which means it does not vary with scale. The different conclusion given by their research might be due to the reason that local deformation and penetration dominated the energy absorption in thick sandwich panels rather than global elastic deformation. Whereas, global elastic deformation plays a significant role in energy absorption for the relative thin composite laminates. 

The present data appear to show that the size of the sample does not have a strong effect on the fiber failure values of maximum load or absorbed energy. The evidence shown in Figure 4 and Figure 6 seem to show this quite clearly. It is possible that the maximum impact force in the larger specimens are underpredicted by linear analysis to a greater degree than in the smaller specimens. Finite element analysis on scale effect of the impact response for GFRP laminates should be carried out in the future work based on the detailed experimental results present in this study. The microstructural effect at higher impact energy can be investigated to fully understand the mechanisms of the scale effect on the impact behavior of GFRP laminates.

## 6. Conclusions

The aim of this work is to evaluate the effects of geometric scale on the impact response, damage and failure of glass fiber composite structures. The scaling effect on the low-velocity impact response of unidirectional GFRP laminates was investigated based on a geometry similar scaling law. A series of impact energies ranges, which can cause damage from barely invisible damage to large scale delamination and initiated penetration, were applied on impacting the GFRP laminates. It was found that the impact response of larger GFRP laminates can be accurate predicated using a simple geometrical similar scaling law when the impact response is elastic. The internal damages modes such as matrix carking and delamination exhibit a dependence on laminate size as would be expected from fracture mechanics considerations. However, the impact force and energy absorption of the ¼ size samples with large scale delamination and penetration, showing great difference to the other three scale size laminates, due to the microstructural effect playing a significant role in the small size sample when significant damages are generated. In practical structural design when significant damages are involved, smaller size samples should be avoided on predicting the impact responses of composite laminates with a very large scale factor. To sum up, these results are still encouraging and suggest that the scaling law has potential to predict the impact response of larger structures.

## Figures and Tables

**Figure 1 materials-12-01319-f001:**
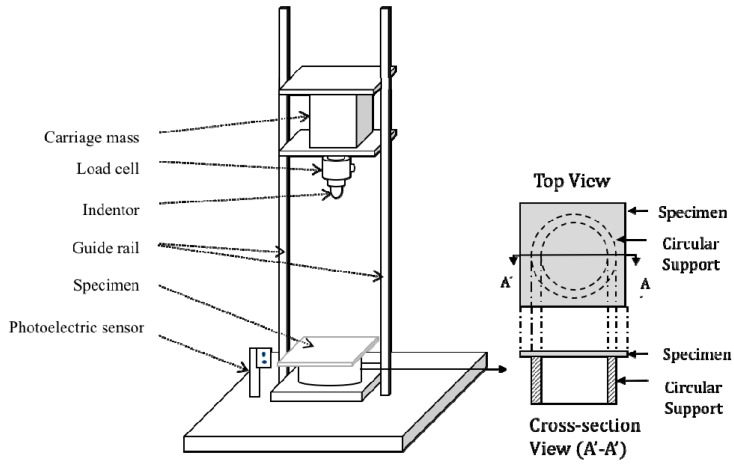
Schematic diagram of the drop weight impact machine and circular support.

**Figure 2 materials-12-01319-f002:**
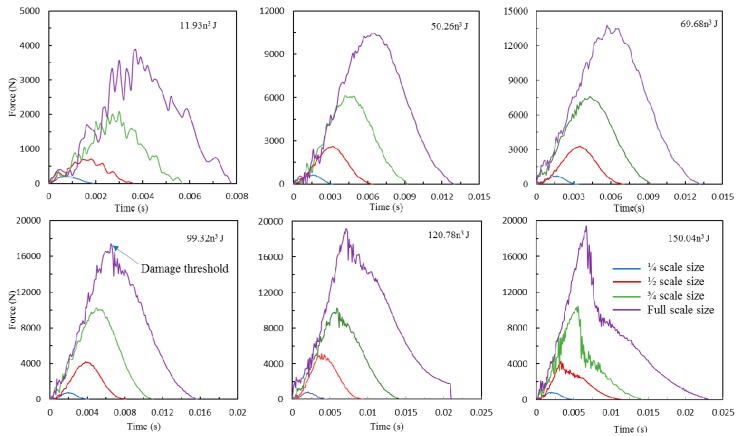
Force vs. time traces of four scale size GFRP laminates following six series of impact energies.

**Figure 3 materials-12-01319-f003:**
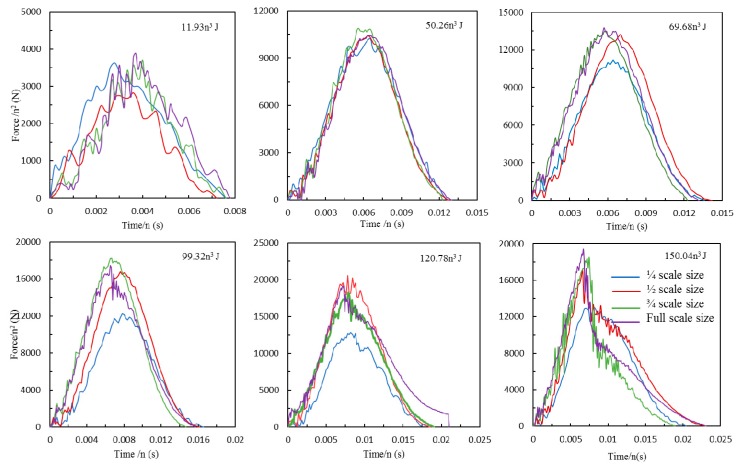
Normalized force vs. time traces of four scale size GFRP laminates following six series of impact energies.

**Figure 4 materials-12-01319-f004:**
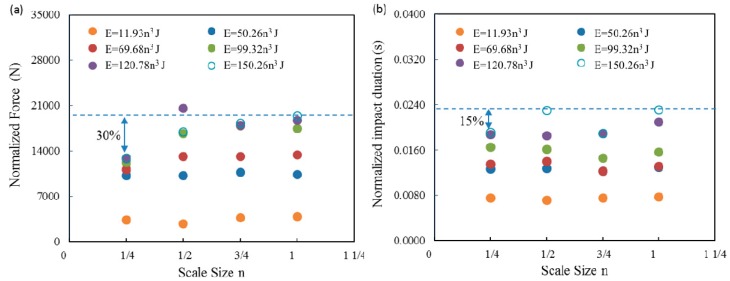
Variation of (**a**) normalized impact force vs. scale size n and (**b**) normalized impact duration vs. scale size n of the GFRP laminates at the six scaled impact energies.

**Figure 5 materials-12-01319-f005:**
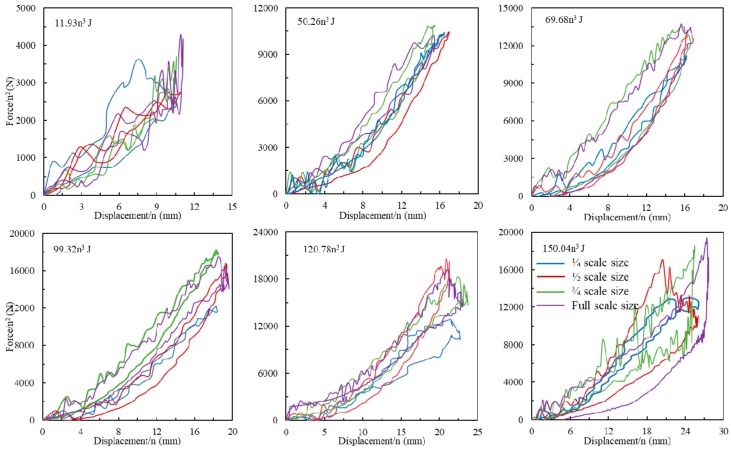
Normalized force vs. displacement traces of four scale size GFRP laminates following six series of impact energies.

**Figure 6 materials-12-01319-f006:**
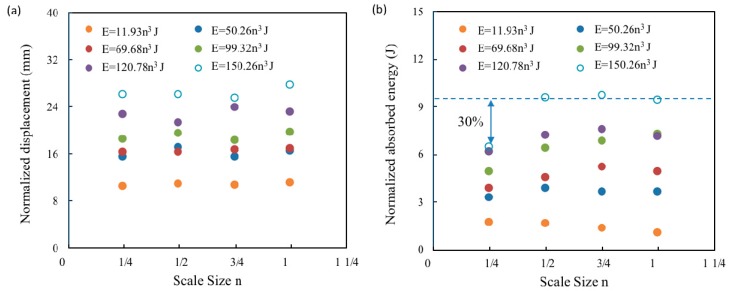
Variation of (**a**) normalized displacement and (**b**) normalized impact energy with scale size n of the GFRP laminates at the six scaled impact energies.

**Figure 7 materials-12-01319-f007:**
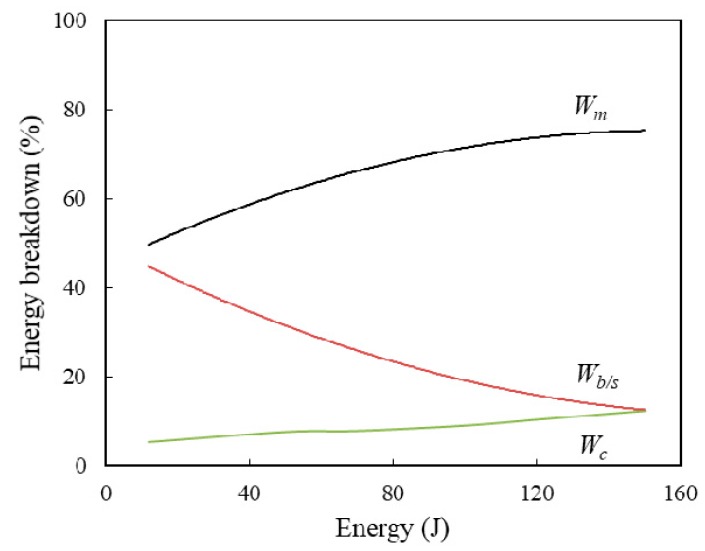
Variation of energy partition profiles for the full-scale size GFRP with different impact energies. *W_m_*, *W_b/s_* and *W_c_* represent the energy absorbed by membrane, bending/shear and contact deformation, respectively.

**Figure 8 materials-12-01319-f008:**
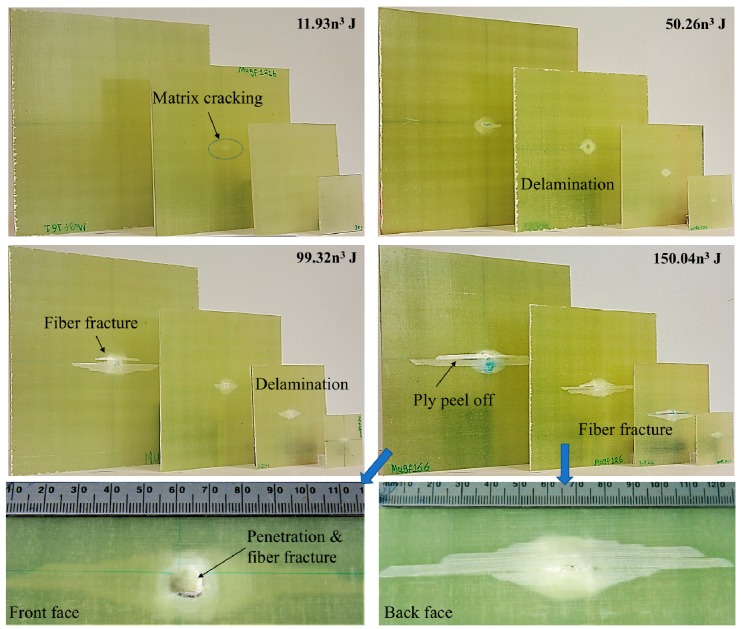
Photograph of the four scale size specimens following impact energies of 11.93n^3^, 50.26n^3^, 99.32n^3^ and 150.26n^3^ J.

**Figure 9 materials-12-01319-f009:**
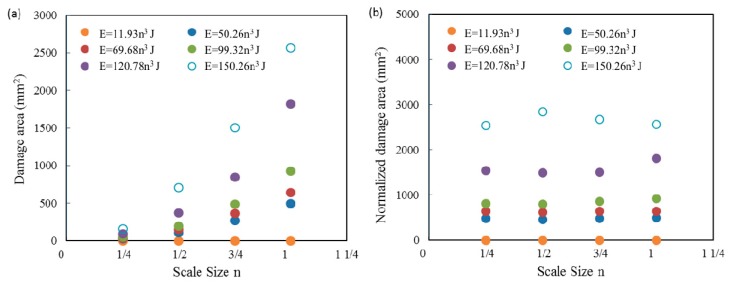
Variation of the (**a**) damage area and (**b**) normalized damage area with scale size n of the GFRP laminates following impact at the six scaled energies.

**Table 1 materials-12-01319-t001:** Scaling of impact response parameters.

Symbol	Parameter	Dimension	Scale Factor
*h*	Specimen Thickness	L	n
*R*	Specimen Radius	L	n
*ρ*	Specimen Density	M L^−3^	1
*υ*	Specimen Poisson’s Ratio	-	1
*E*	Specimen Modulus	M T^−2^ L^−1^	1
*m_i_*	Impactor Mass	M	n^3^
*V_i_*	Impactor Velocity	L T^−1^	1
*ρ_i_*	Impactor Density	M L^−1^	1
*υ_i_*	Impactor Poisson’s Ratio	-	1
*E_i_*	Impactor Modulus	M T^−2^ L^−1^	1
*t*	Impact Time	T	n
*δ*	Impact Deflection	L	n
*P*	Impact Force	M T^−2^ L	n^2^

**Table 2 materials-12-01319-t002:** Dimensions of the scaled glass fiber reinforced plastic (GFRP) plates and impact testing conditions.

Scale Factor λ	Edge Length (mm)	Avg. Thick (mm)	No. of Plies	Impact Height (mm)	Indenter Diameter (mm)	Support Diameter (mm)
¼	65	0.95	4	500	5	50
½	130	1.95	8	500	10	100
¾	195	3.0	12	500	15	150
1	260	4.0	16	500	20	200

**Table 3 materials-12-01319-t003:** Summary of the impact tests parameters and their dependency on the scaling factor.

Scale Size	11.93n^3^ J		50.26n^3^ J		69.68n^3^ J		99.32n^3^ J		120.78n^3^ J		150.04n^3^ J	
	*m_i_* (kg)	*W* (J)	*m_i_* (kg)	*W* (J)	*m_i_* (kg)	*W* (J)	*m_i_* (kg)	*W* (J)	*m_i_* (kg)	*W* (J)	*m_i_* (kg)	*W* (J)
¼	0.04	0.19	0.16	0.79	0.22	1.09	0.32	1.56	0.39	1.896	0.48	2.34
½	0.31	1.5	1.28	6.26	1.79	8.76	2.55	12.5	3.07	15.07	3.82	18.76
¾	1.03	5.03	4.30	21.09	6.02	29.55	8.57	42.05	10.38	50.91	12.91	63.32
1	2.43	11.93	10.25	50.26	14.21	69.68	20.25	99.32	24.62	120.78	30.59	150.04

**m_i_* represents impactor mass, *W* represents impact energy.

**Table 4 materials-12-01319-t004:** Variation of effective stiffness for four scale size GFPR laminates.

Scale Size	1/4	1/2	3/4	1
*K_b_*	7.04 × 10^4^	1.41 × 10^5^	2.29 × 10^5^	3.05 × 10^5^
*K_s_*	3.26 × 10^7^	2.76 × 10^7^	4.24 × 10^7^	5.64 × 10^7^
*K_b/s_*	7.02 × 10^4^	1.41 × 10^5^	2.28 × 10^5^	3.03 × 10^5^
*K_m_*	2.12 × 10^10^	1.01 × 10^10^	6.92 × 10^9^	5.19 × 10^9^

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
