# Peer review of "Scale Effect on Impact Performance of Unidirectional Glass Fiber Reinforced Epoxy Composite Laminates"

_materials, 2019, doi:10.3390/ma12081319_

Round 1
Reviewer 1 Report
Well written paper on the interesting and important from practical point of view thopic. The research methodology is good. The results are clearly presented and the discussion is exhaustive. The conclusions are short but give the a good picture of the obtained most important test results. The bibliopgraphy is quite extensive and actual with some papers from 2019 year. The final evaluation of the paper is good and it can be published in Materials Journal in the present form.
Author Response
Authors would like to thank the precious advices from the reviewer. This paper has been carefully modified according to the reviewer's advices.
Reviewer 2 Report
Title: Scale effect on impact performance of unidirectional glass fiber reinforced epoxy composite laminates
Authors: Yiou Shen and colleagues
Overall assessment:
The authors investigated the behaviors of a GFRP under various impact loadings. Although the topic adopted by the authors seems to be interesting, the descriptions are unfortunately too casual and ambiguous. Therefore, I have a great concern that many readers cannot appreciate the essence of this study appropriately due to the casualties and ambiguities. I recommend the authors to withdraw the submission in this occasion, and prepare a new version in which the ambiguities and casualties are thoroughly reduced by objective readings.
Followings are the specific comments which may be helpful for resubmission:
Specific comments:
1. When seeing the descriptions in Materials and Methods section alone, it is difficult to understand the testing conditions precisely. For example, it seems to me that four-points of the square specimen may be supported using the apparatuses listed in Table 2, but the details are entirely missing.
If the test method is determined in some major standard, the authors should quote the standard in the manuscript. Otherwise, a photograph and/or diagram for the experiment should be demonstrated.
2. The descriptions on the experimental methods are too terse. For example, there are no descriptions how to measure the displacement. The authors must elaborate the descriptions on the experimental methods more in details.
3. If the scale factor is represented using “n”, the number of the ply of the GFRP laminates should be represented using the other character.
4. The impact energy should not be represented using “E” in Table 3 because there is a concern that “E” is regarded wrongly as the specimen modulus listed in Table 1.
5. It is very difficult to understand what Table 3 represents immediately. For example, the value of impact energy of 12n^3 J is not 12 J but 11.93 J. Such a discrepancy confuses the readers. To enhance the precise understandings of the readers, the authors must elaborate the presentation.
6. It seems to me that the results shown in Figs. 1 and 2 are similar with each other. I think that the results in Fig. 1 are not the force-time traces but the normalized force-normalized time traces. The authors must check it.
7. The definitions of the dashed lines in Figs. 3 and 5 should be demonstrated in the captions or in the figures themselves.
8. It is questionable why Table 2 does not contain the Ks values.
9. There are no descriptions how to derive the relations shown in Fig. 6. Additionally, the definitions of Em, Eb/s, and Ec are entirely missing.
10 The label of the vertical axis of Fig. 8b is mistyped as “Normalized damage are”.
Recommendation
I have to provide a negative evaluation and recommend the authors to resubmit the manuscript because of many casualties and ambiguities in the manuscript.
Author Response
The authors would like to thank the precious advice from the reviewer. This paper has been carefully modified according to the reviewer's advices. Please find the attachment about the responses.

Reviewer 3 Report
The article analysis scale glass fiber reinforced epoxy composite laminated and scale effect on impact performance. The article is nicely written and is under the scope of a journal. However, I have some remarks which I hope will be addressed:
Abstract part must present the main results obtained by the research.
Introduction section should be updated with discussion of newer references.
Scaling theory part consists of a reference [4] which is marked in red. Why?
Figures 1,2,5,6,8 have strange frames. please remove them.
Results and discussion part does not contain any discussion with other authors works and their obtained results. Please improve this part.
What about standard deviations of the obtained results in Tables? How many samples were used in order to obtain the average value? If more than one, please indicate standard deviations or confidence intervals.
Author Response
The authors would like to thank the precious advices from the reviewer. This paper has been carefully modified according to the reviewer's advices. Please find the attachment of the responses.

Round 2
Reviewer 2 Report
The revisions are adequately conducted; therefore, I'd like to recommend the revised version to be published.
Reviewer 3 Report
Authors have addressed all my remarks; therefore, I suggest acceptance of the article.